# Activation of c-Jun by human cytomegalovirus UL42 through JNK activation

**Tetsuo Koshizuka** [ID]*, **Naoki Inoue**

Microbiology and Immunology, Gifu Pharmaceutical University, Gifu, Japan

* koshizuka-te@gifu-pu.ac.jp

## Abstract

c-Jun is a major component of the AP-1 transactivator complex. In this report, we demonstrated that AP-1 was activated by the expression of UL42, a human cytomegalovirus-encoded membrane protein that has two PPXY (PY) motifs and a C-terminal transmembrane domain (TMD). Although UL42 interacts with Itch, an ubiquitin E3 ligase, through the PY motifs, UL42 phosphorylated c-Jun and c-Jun N-terminal kinase (JNK) in the absence of any interaction with Itch. Experiments using mutated versions of UL42 suggest the importance of the carboxyl half (a.a. 52–124) of UL42 for the activation of the JNK signaling, while C-terminal TMD alone is not sufficient. Thus, we hypothesize that UL42 plays a role in the activation of JNK signaling in HCMV-infected cells. (118 words).

## Introduction

The proto-oncogene c-Jun, one of the most studied transactivator proteins, is a major component of the heterodimeric AP-1 transcription factor family [1]. Activated c-Jun is transported into the nucleus, where it forms the AP-1 heterodimer complex and binds to promoter regions of target genes. Phosphorylation of c-Jun at Serine 63 (Ser63) and Ser73 by c-Jun N-terminal kinase (JNK) regulates c-Jun transcription activities [2, 3]. The c-Jun/JNK pathway is activated by various extracellular stimuli, including infection, inflammation, oxidative stress, DNA damage, osmotic stress, and cytoskeletal changes [4]. As JNK is a key component of the innate immunity pathways, pathogens have developed strategies to modulate the JNK signaling events [4]. While suppression of JNK signaling has some advantages to many pathogens, other pathogens activate the JNK pathway. For example, Epstein-Barr virus (EBV) LMP1 activates JNK through TRAF signaling [5]. Human cytomegalovirus (HCMV) IE1 activates the phosphorylation of c-Jun [6]. Further, the activation of JNK is essential for effective viral protein expression and replication in varicella-zoster virus-infected neuronal cells [7]. Therefore, the regulation of c-Jun/JNK signaling by viral proteins is important for the replication of some viruses.

The UL42 gene product of HCMV is a membrane protein that contains two PPXY (PY) motifs to interact with Itch, a member of the ubiquitin E3 ligase Nedd4 family [8]. UL42 and its alpha- and beta-herpesvirus homologs share a number of conserved structures including the PY motifs in their N-terminal domain and the C-terminal transmembrane domain (TMD), but the function of other domains remains to be elucidated [9–11]. All these homologs interact with Itch through their PY motifs. As Itch ubiquitinates various substrates, it plays multiple roles in signal transduction, intracellular trafficking, cell survival and immune responses [12].

**Data Availability Statement:** All relevant data are within the paper and its Supporting Information files.

**Funding:** This work was partly supported by the Grants-in-Aid for Scientific Research from the Japan Society for the Promotion of Science (JSPS

KAKENHI 17K10185) to T.K. and a grant from the Takeda Science Foundation to T.K.

**Competing interests:** The authors have declared that no competing interests exist.

Indeed, Itch is involved in the negative regulation of c-Jun/JNK signaling through ubiqutination of c-Jun [13]. Fu and colleagues have recently reported that UL42 inhibits DNA binding, oligomerization and enzymatic activity of cyclic GMP-AMP synthase to antagonize innate antiviral responses in a Nedd4 family- independent manner [14].

In the present study, we investigated whether UL42 regulated c-Jun activation through its interaction with Itch. For this purpose, we performed mapping of the UL42 functional domains for AP-1 transcriptional activation, nuclear localization of c-Jun, and phosphorylation of c-Jun and JNK. Unexpectedly, we found that UL42 activated c-Jun in an Itch-independent manner. Thus, UL42 has the ability to regulate JNK signaling among HCMV-encoded proteins.

## Materials and methods

### Cells

The HEK293T cells (RIKEN Cell Bank, Tsukuba, Japan) were cultured in Dulbecco's minimum essential medium (DMEM) supplemented with 10% fetal bovine serum (FBS), 100U/ml penicillin and 100U/ml streptomycin. Immortalized human fibroblasts (hTERT-BJ1) were cultured in DMEM-medium 199 (4:1) supplemented with 10% FBS, 2 mM L-glutamine, 1 mM sodium pyruvate, 100 U/ml penicillin, and 100 μg/ml streptomycin.

### Plasmids and transfection

pCAGGS-HAUL42WT, pCAGGS expressing wild-type UL42 with an HA-tag at the N-terminus, was described previously [8, 15]. Primers P1-13 used in this report were shown in Supporting Information (S1 Table). pCAGGS-HAUL42ΔN and -HAUL42ΔI, pCAGGS expressing HA-tagged UL42 lacking the amino acid (a.a.) 1–50 and 51–86 regions, respectively (Fig 1A), were constructed by the inverse PCR-based method using pCAGGS-HAUL42WT as a template along with primer pairs P1 and P2 for ΔN and P3 and P4 for ΔI, respectively.

A PCR-amplified fragment encoding the UL42 open reading frame using with primers P5 and P6 was inserted between the BamHI and XhoI sites of pEGFP-C1 (Takara Bio, Shiga, Japan) to construct pEGFP-UL42WT. pEGFP-UL42AY, a PY motif-disrupted (PPXY to AAXY alteration) mutant, was constructed by the QuickChange site-directed mutagenesis (Agilent technologies, St Clara, CA) of pEGFP-UL42WT using primers P8-P11.

The C-terminal TMD of UL42 was amplified by PCR using primers P4 and P7 and inserted between the BglII and SalI sites of pEGFP-C1, generating pEGFP-UL42Ct. Integrities of all inserts were confirmed by DNA sequencing. All plasmids used were purified with a Qiagen plasmid plus midi kit (Qiagen, Venlo, Netherlands). HEK293T cells were transfected with the indicated plasmids using ScreenFect A (Fuji-Film Wako Pure Chemical, Osaka, Japan).

### Luciferase assay

The control and reporter plasmids, 0.01 μg/well of pRL-TK (Promega, Madison, WI) and 0.1μg/well of pAP1(PMA)-TA-Luc, which contains the firefly luciferase gene under the control of a minimal promoter with multiple copies of the AP-1 enhancer elements (Takara Bio), and 0.1 μg/well of UL42 expression plasmids were transfected to HEK293T cells in 96-well plates. At 48h post-transfection, the luciferase activity of the cells was analyzed with a Dual-Glo Luciferase assay kit (Promega). The ratios of firefly luciferase activities to Renilla luciferase activities were obtained in triplicated wells in each experiment.

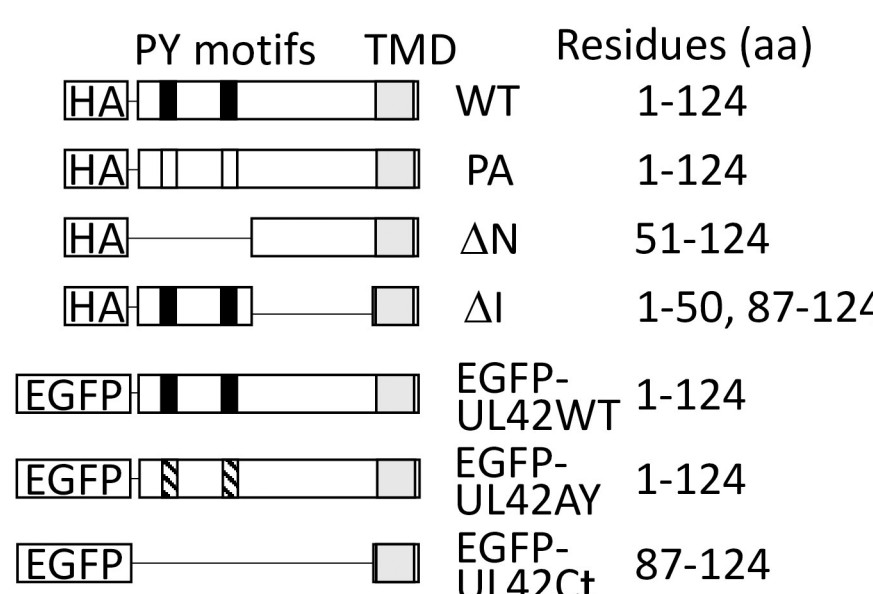

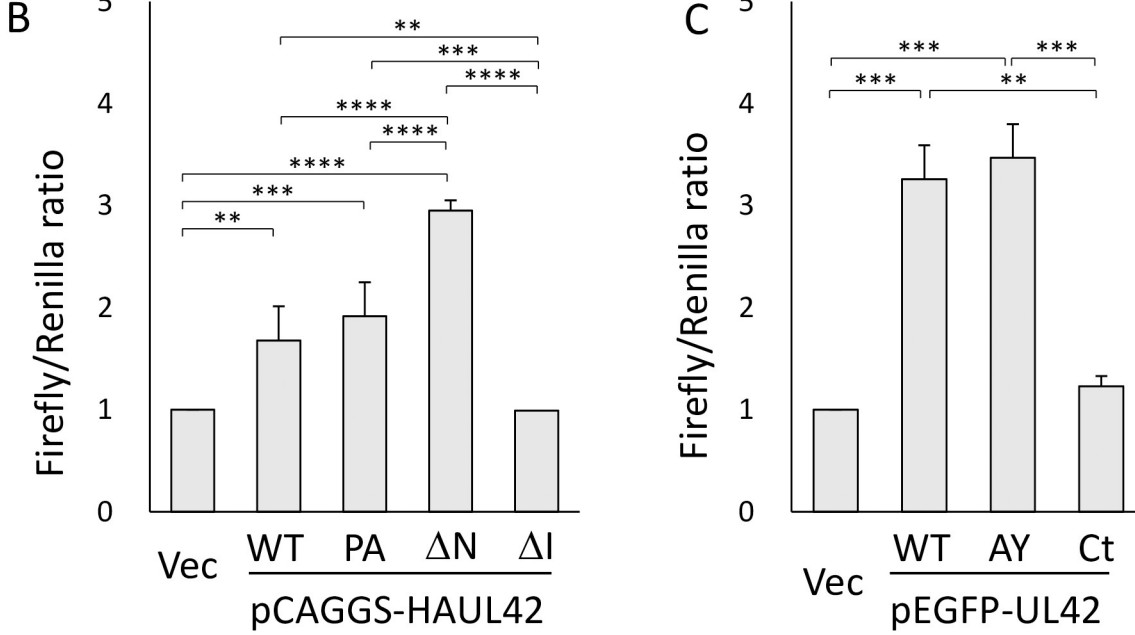

**Fig 1. UL42 mutant constructs and their ability to activate AP-1-dependent transcription.** A. HCMV UL42 has two PY motifs (closed boxes) and a C-terminal transmembrane domain (TMD; a closed box in gray). In the PA and AY mutants, the PPXY sequences were substituted to PPXA (open boxes) and AAXY (hatched boxes), respectively. UL42 and its mutated forms were fused with HA or EGFP tags at the N-terminus of UL42. Residue numbers are based on the position in wild-type (WT) UL42. B and C. Indicated plasmid expressing HA-tagged (B) or EGFP-tagged (C) UL42 mutants and the luciferase reporter and control plasmids, pAP1(PMA)-TA-Luc and pRL-TK, were transfected into HEK293T cells. Means ± SEMs of the ratios of firefly luciferase activities to Renilla luciferase activities obtained in three independent experiments (S1 Fig) are shown. The p-values were determined by the one-way ANOVA followed by the Tukey's multiple comparison test. **p<0.01, ***p<0.005, ****p<0.001. (B) pCAGGS (Vec), pCAGGS-HAUL42WT (WT), -HAUL42PA (PA), -HAUL42ΔN (ΔN), and -HAUL42ΔI (ΔI). (C) pEGFP-C1 (Vec), -UL42WT (WT), -UL42AY (AY), and -UL42Ct (Ct).

## Antibodies

The anti-HA rabbit polyclonal antibody, and anti-GFP monoclonal antibody (dilution ratio 1:1000) (MBL, Nagoya, Japan), anti-c-Jun (dilution ratio 1:1000), anti-phospho-c-Jun (S63) (dilution ratio 1:1000), and anti-Itch monoclonal antibodies (dilution ratio 1:1000) (BD Bioscience, Franklin Lakes, NJ), anti-JNK and phospho-JNK rabbit polyclonal antibodies (dilution ratio 1:250)(Cell Signaling Technology, Danvers, MA), and anti-actin monoclonal antibody (dilution ratio 1:5000) (Merck, Darmstadt, Germany) were purchased as indicated. Anti-UL42 rabbit polyclonal antibody was raised against GST-UL42 (dilution ratio 1:2000). Peroxidase-conjugated anti-mouse or anti-rabbit IgG antibodies (dilution ratio 1:2000 and 1:5000, respectively) (GE healthcare, Chicago, IL), and Alexa Fluor 488-, 594- or 647-conjugated antibodies (Thermo Fisher Scientific, Waltham, MA) were used as the secondary antibodies.

## Immunoblots and immunofluorescence analysis

Immunoblotting analyses were performed essentially as described elsewhere [9]. In brief, HEK293T cells were transfected with 0.6 μg/well of plasmids, cultured for 48h and lysed in the Laemmli's sample buffer. After boiling at 98 ˚C for 5 min and brief sonication, cell lysates were separated on sodium dodecyl sulfate-poly-acrylamide gels and transferred to PVDF membranes (Millipore). After blocking with PBS-T (0.05% Tween 20 in PBS) containing 5% skim milk (Fuji-Film Wako Pure Chemical), the membranes were washed three times with PBS-T and incubated with primary antibodies diluted in PBS-T containing 1% bovine serum albumin (BSA) at 4 ˚C for overnight. After washing three times with PBS-T, membranes were incubated with peroxidase-conjugated secondary antibodies diluted into PBS-T containing 5% skim-milk at room temperature for 3 h, and then washed three times with PBS-T. After reaction with Immunostar LD reagent (Fuji-Film Wako Pure Chemical), signals were detected in ChemiDoc system (BioRad Laboratories, Hercules, CA).

Cells were grown on coverslips and fixed with 4% paraformaldehyde in PBS at 48 h post-transfection. The cells were permeabilized with PBS containing 0.05% Triton X-100 and stained with DAPI (Thermo Fisher Scientific) and the indicated antibodies at room temperature for 30 min. Samples were mounted on slide glass with antifade (Thermo Fisher Scientific) and analyzed with confocal microscopy (LSM700, Carl Zeiss, Oberkochen, Germany). Data capture was done under conditions identical among series of the samples.

## Recombinant HCMV strains

The recombinant viruses were constructed using the two-step Red-mediated mutagenesis [16]. To recover the epithelial and endothelial cell tropisms of HCMV strain Towne, the UL130 gene was repaired as described previously [17] to generate TowBACdTT-WT genome. The UL42 open reading frame was deleted from TowBACdTT-WT genome to construct TowBACdTT-ΔUL42 as described previously [8]. Briefly, a DNA fragment containing kanamycin resistant gene (Km^r) and I-SceI site was amplified with primers P12 and P13 and inserted into pCAGGS-UL42WT and pCAGGS-UL42PA. I-SceI-Km^r-UL42 fragments were amplified with primers P13 and P14, inserted into TowBACdTT-ΔUL42 genome, and Km^r sequence was removed to generate TowBACdTT-UL42R and TowBACdTT-UL42PA. Successful recombination was confirmed by PCR, Southern blotting and DNA sequencing. Those BAC genomes were purified using a Nucleobond BAC 100 kit (TaKaRa Bio) and transfected to human fibroblasts to reconstitute infectious recombinant viruses.

## Statistical analysis

Data of luciferase assays and cell counting were based on three independent experiments, and each set of experimental results was analyzed statistically with GraphPad PRISM software ver.8.3 (GraphPad Software, San Diego, CA). The luciferase data obtained in triplicated wells were analyzed with the one-way ANOVA test followed by the Tukey's multiple comparison test. Immunofluorescence assay data were digitalized with Photoshop software (Adobe systems, San Jose, CA), and the percentages of nuclear c-Jun-positive cells among the cells expressing UL42 derivatives or EGFP were obtained in three random fields in each experiment. Then, the percentages obtained in three independent experiments were analyzed with the one-way ANOVA test followed by the Tukey's multiple comparison test.

## Results

### AP-1 signaling activation by UL42 expression

To measure the AP-1-dependent transcriptional activities, HEK293T cells were transfected with a UL42-expressing plasmid and a reporter plasmid, pAP1(PMA)-TA-Luc, for a luciferase assay. Expression of UL42 wild type (WT) enhanced luciferase activities (Fig 1B). Expression of UL42 mutant protein UL42PA, which contains alterations of the two PPXY (PY) motifs to the PPXA (PA) motifs to abolish binding activity to the Nedd4 family, yielded the same luciferase activity as UL42WT. We constructed several deletion mutants of UL42 (Fig 1A) to identify the domain responsible for AP-1 transcriptional activation. The deletion of the N-terminal region (ΔN) induced luciferase activity as that of UL42WT, but the deletion of the internal region (ΔI) showed reduced activity, suggesting the requirement of the C-terminal TMD for the AP-1 signaling. To confirm two observations described above, *i.e.* no involvement of the Nedd4 family E3 ligases and the requirement of the C-terminal TMD for induction of the AP-1 signaling, we constructed EGFP-UL42AY and EGFP-UL42Ct for the respective purposes. EGFP-tagged UL42 derivatives EGFP-UL42WT and -UL42AY but not EGFP-UL42Ct which contains only C-terminal TMD activated AP-1 signaling (Fig 1C), indicating that the C-terminal half of UL42 rather than the PY motifs is essential for the activation of AP-1-dependent transcription.

### The nuclear accumulation of c-Jun by UL42 expression

As it is well known that the activated form of c-Jun is translocated to the nucleus for AP-1 formation, we examined the intracellular localization of c-Jun in UL42-expressing cells. As shown in Fig 2, transfection with a plasmid expressing UL42WT, PA orΔN increased the numbers of c-Jun-positive nuclei in comparison with transfection with a control vector plasmid. C-Jun was only weakly detectable in cells expressing UL42ΔI. As reported previously, UL42WT and UL42PA were localized in cytoplasmic membranous structures [8]. The mutant proteins containing the C-terminal TMD were localized in the cytoplasmic structures and such localizing patterns resembled that of UL42WT. In addition, the percentages of nuclear c-Jun-positive cells in cells expressing HA-tagged UL42 derivatives were evaluated as described in Materials and Methods. C-Jun was highly accumulated in the nuclei of cells expressing HA-tagged UL42WT, UL42PA and UL42ΔN but not those expressing HA-tagged UL42ΔI (Fig 2A), and the defect of UL42ΔI was statistically significant (Fig 2B).

To confirm the results based on HA-tagged UL42 derivatives, we evaluated the nuclear accumulation of c-Jun using EGFP-tagged UL42 derivatives (Fig 3A and 3B). EGFP-UL42WT was localized in the cytoplasmic membranous structures as HA-tagged UL42WT (S2A and S2B Fig). Although the subcellular localization of EGFP-UL42AY was similar to that of

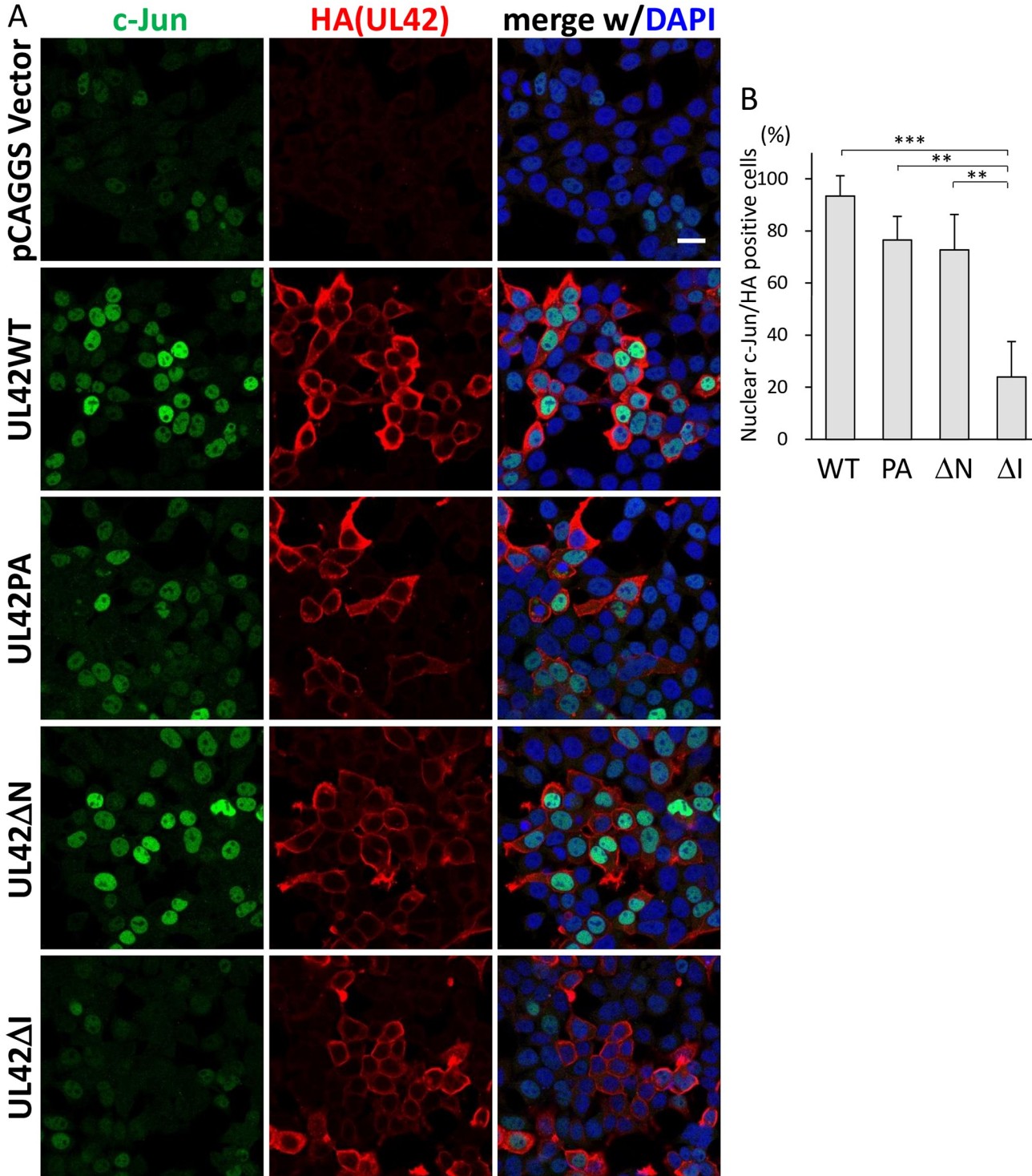

**Fig 2. Intracellular localization of HA-tagged UL42 mutants and c-Jun.** A. Detection of c-Jun and HA-tagged UL42 derivatives. The HEK293T cells were transfected with pCAGGS, pCAGGS-HAUL42WT (WT), -HAUL42PA (PA), -HAUL42ΔN (ΔN), or -HAUL42ΔI (ΔI), and reacted with anti-c-Jun and anti-HA antibodies, and then with Alexa Fluor 488- and 594-conjugated secondary antibodies for c-Jun (shown in green) and UL42 (in red). The nuclei of the cells were stained with DAPI (blue). B. The percentages of nuclear c-Jun-positive cells among UL42 or its derivative expressing cells. Mean ± SEM of the percentages obtained in three independent experiments are shown. Statistical differences were determined by the one-way ANOVA followed by the Tukey's multiple comparison test. **p<0.01, ***p<0.005.

EGFP-UL42WT, EGFP-UL42Ct was detected in a mesh-like localization pattern in the cytoplasm (S2B Fig), which is the characteristic of ER localization. As shown in Fig 3A and 3B, the expression of EGFP-UL42WT and -UL42AY but not of EGFP-UL42Ct induced the nuclear localization of c-Jun in HEK293T cells. The percentages of nuclear c-Jun-positive cells were increased by expression of EGFP-UL42WT and -UL42AY but not of EGFP-UL42Ct (Fig 3B). These results indicate that the C-terminal TMD alone was not sufficient for the nuclear accumulation of c-Jun.

## JNK and c-Jun phosphorylation by UL42 expression

The phosphorylation status of c-Jun, JNK and Itch were analyzed by immunoblotting using lysates of cells expressing UL42 or its mutated forms (Fig 4). As shown in Fig 4A, UL42ΔN could be more sensitive to proteolytic cleavage, as 17 kDa and 8 kDa bands were detected in addition to the 24 kDa full-length product. As reported previously [8], the amount of Itch was decreased by the expression of UL42WT but not of UL42PA. The Ser63 residue of c-Jun was phosphorylated by the expression of UL42WT and UL42PA (Fig 4A). The phosphorylation was further increased by UL42ΔN expression. In addition, JNK was highly phosphorylated by UL42WT, PA and ΔN. In contrast, ΔI only weakly induced the phosphorylation of c-Jun and JNK, while ΔI, which possesses the PPXY motifs, significantly decreased Itch amount. C-Jun and JNK were phosphorylated by the expression of EGFP-UL42WT and -UL42AY but not by EGFP-UL42Ct.

In spite of the evident effect of UL42 on c-Jun in transfection assays, the phosphorylation status of c-Jun was not affected by UL42 expression in HCMV-infected cells (S3 Fig).

## Discussion

Our results revealed that HCMV UL42 induced AP-1 signaling by the activation of the c-Jun/JNK pathway. UL42 expression increased the phosphorylation of JNK as well as that of c-Jun Ser63, one of two JNK-mediated phosphorylation sites required for the promotion of c-Jun transactivation activity [1]. UL42 reduced Itch amount, a member of the ubiquitin E3 ligase Nedd4 family [8]. Itch belongs to the negative feedback mechanism of JNK, as activated Itch ubiquitinates c-Jun to induce their degradation [12, 13]. Importantly, however, the lack of the PY motif for binding to Itch in UL42PA and ΔN mutants did not decrease JNK signaling, indicating that Itch is not involved in the UL42-mediated JNK activation. Although the phosphorylation levels of c-Jun seems weak in EGFP-UL42WT and -AY expressing cells as compared to those in cells expressing HA-tagged UL42WT and PA, the phosphorylation of JNK was significantly increased in EGFP-UP42WT and -AY expressing cells (Fig 3B). We assume that steric hindrance due to EGFP fusion reduced the levels partially. In fact, the results of luciferase assay (Fig 1C) and nuclear translocation of c-Jun (Fig 3) supported the notion that the phosphorylation of c-Jun and activation of AP-1 signaling occurred by expression of EGFP-UL42WT and -AY but not -Ct. As JNK is a member of MAPK, which is activated by various stimuli, including infection, inflammation, oxidative stress, DNA damage, stimulation, if at all, of these signaling pathways.

The results of our domain mapping experiments suggest that the a.a. 52–86 region of UL42 is responsible for the c-Jun nuclear localization, as the a.a. 52–124 region, but not the C-terminal TMD itself, activated AP-1 signaling (Fig 1), re-localized c-Jun to the nucleus (Figs 2 and 3), and c-Jun phosphorylation (Fig 4). Although EGFP-UL42Ct expressed a lower level of its UL42 product than the other constructs in both immunofluorescence and immunoblotting assays, c-Jun was not re-localized to the nuclei in the EGFP-UL42Ct expressing cells, suggesting that the C-terminus of UL42 was not responsible for the activation of c-Jun. It is unlikely

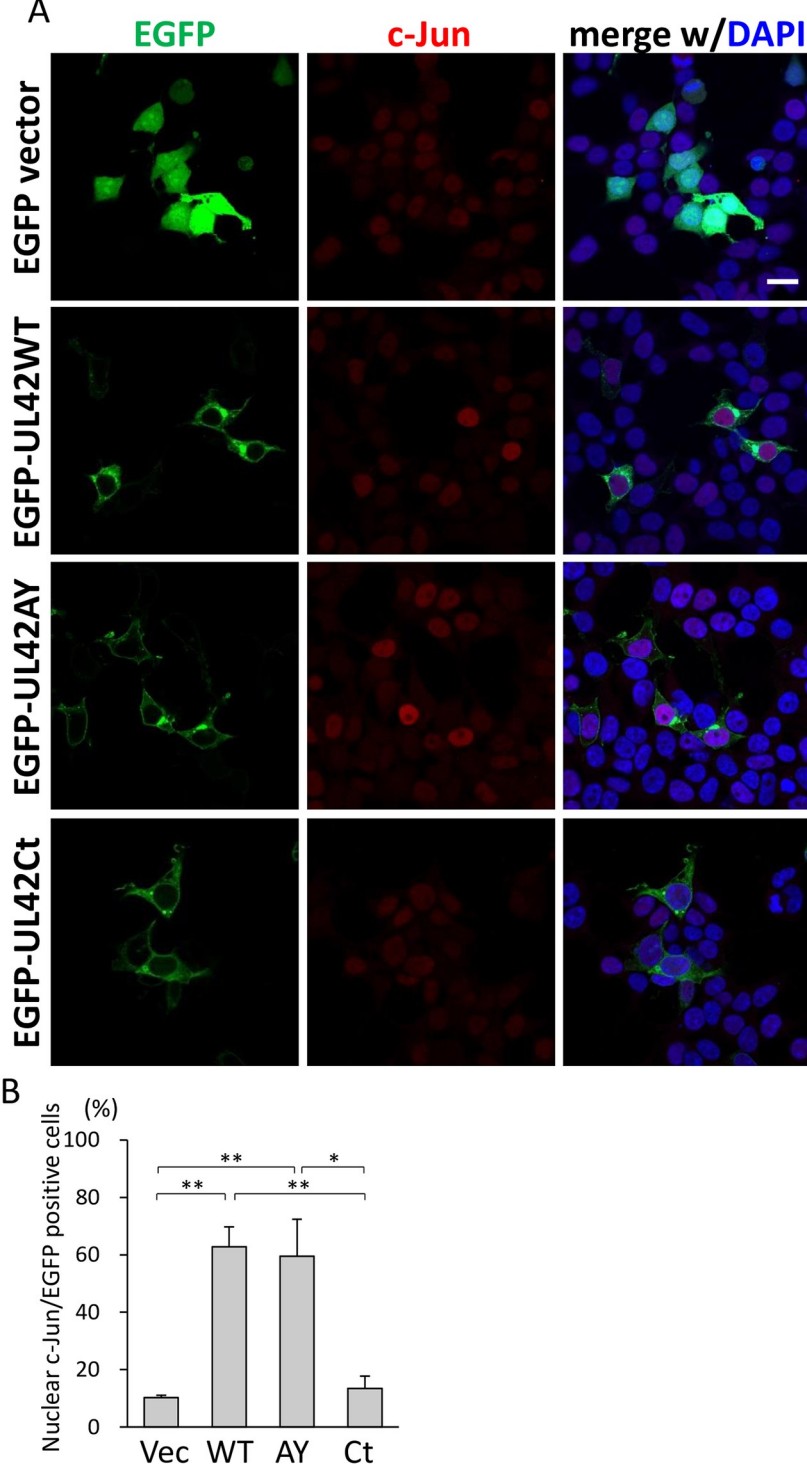

**Fig 3. Intracellular localization of EGFP-tagged UL42 mutants and c-Jun.** A. The plasmids expressing the indicated proteins were transfected to HEK293T cells. The cells were reacted with anti-c-Jun and then with Alexa Fluor 647-conjugated secondary antibody (shown in red). EGFP fluorescence and nuclei staining with DAPI are shown in green and blue, respectively. B. The percentages of nuclear c-Jun-positive cells among cells expressing EGFP (Vec), EGFP-UL42WT (WT), -UL42AY (AY), or -UL42Ct (Ct). Mean ± SEM and statistical differences are shown as described in the legend for Fig 2B. $^*p < 0.05$, $^{**}p < 0.01$.

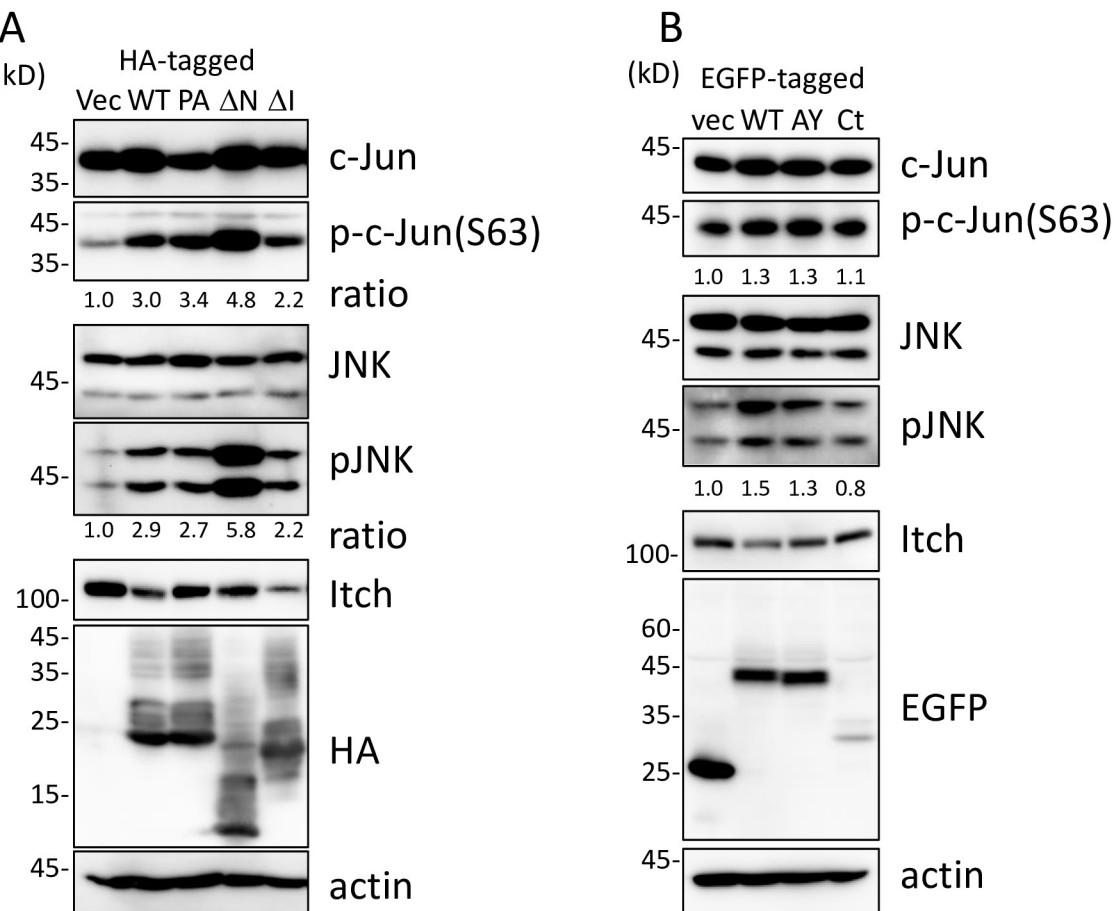

**Fig 4. Detection of phosphorylated JNK, c-Jun and Itch in cells expressing UL42 derivatives.** A. Lysates of HEK293T cells transfected with pCAGGS (Vec), pCAGGS-HAUL42WT (WT), -HAUL42PA (PA), -HAUL42ΔN (ΔN), or -HAUL42ΔI (ΔI) were analyzed by immunoblotting using antibodies for the detection of the indicated forms of proteins. Ratios of band intensities between the forms of the indicated protein without and with phosphorylation were analyzed with NIH imageJ software, and indicated beneath the panels. B. Lysates of HEK293T cells transfected with pEGFP-C1 (Vec), pEGFP-UL42WT (WT), -UL42AY (AY), or -UL42Ct (Ct) were analyzed by immunoblotting using antibodies for the detection of the indicated forms of proteins.

that the expression levels of EGFP-UL42 variants affected the nuclear localization and activation of c-Jun, because nuclear accumulation of c-Jun was observed in all cells expressing HA-UL42WT, -UL42PA or -UL42ΔN but not those expressing HA-UL42ΔI, while the expression levels of HA-tagged proteins in individual cells varied significantly. The N-terminal region is thought to contain a regulatory domain for the control of JNK signaling, as the deletion of the N-terminal region enhanced AP-1 activity more than that of UL42WT. Interestingly, the expression of the UL42ΔI mutant did not increase c-Jun nuclear import but induced phosphorylation of c-Jun to the same degree as UL42WT, which is consistent with a previous report that the nuclear import of c-Jun is independent of its phosphorylation [18]. Further investigation will be needed to elucidate the precise functional domains of UL42.

In contrast to outcomes in transient transfection assays which performed in HEK293T cells, the lack of UL42 did not affect the phosphorylation status of c-Jun in HCMV-infected fibroblasts (S3 Fig). As we did not examine the effects of UL42 constructs in fibroblasts due to a low transfection efficiency, one potential explanation of the discrepancy would be a cell-type specific function of UL42. However, it is more plausible that the presence of multiple mechanisms of the c-Jun phosphorylation in HCMV-infected cells results in the discrepancy. Indeed, the activation of c-

Jun/JNK signaling occurs immediately after HCMV infection and a JNK inhibitor, SP600125, inhibits HCMV replication by the suppression of immediate-early gene expression [19]. HCMV encodes many c-Jun/JNK signal-modulating proteins, including IE1 and UL38 [6, 20]. IE1, a transactivator encoded by HCMV, promotes phosphorylation of c-Jun through a cellular protein kinase [6, 21]. On the other hand, another HCMV-encoded protein, UL38, which is classified as an early gene product [22], reduces JNK phosphorylation to suppress ER stress-induced cell death [20]. Previous studies demonstrated that UL42 was classified as a dispensable gene [8, 23], which is consistent with the notion that HCMV has several c-Jun modifying genes. As UL42 is known to be expressed at least 1 day post-infection in HCMV-infected fibroblasts [8], we hypothesize that UL42 would contribute to regulate c-Jun/JNK signaling in some manner during the early phase of infection with some HCMV encoded protein(s).

In conclusion, UL42 activated c-Jun/JNK signaling in an Itch-independent manner. The results presented herein indicate that the a.a. 52–86 regions of UL42 is responsible for c-Jun nuclear translocation. This region is important for induction of AP-1 signaling and JNK activation. In the future, it would be interesting to see the Nedd4 family-independent inhibitory effect of UL42 on cyclic GMP-AMP synthase [14]. Further research is still required, however, to elucidate the precise mechanisms of the UL42-mediated activation of c-Jun/JNK signaling.

## Supporting information

**S1 Data.**
(PDF)

**S1 Table. Primer list.**
(DOCX)

**S1 Fig. Luciferase assay results of UL42 derivatives.** The luciferase assay results of three independent experiments are shown. A plasmid expressing the indicated UL42 mutant tagged with HA (A) or with EGFP (B), the luciferase reporter plasmid pAP1(PMA)-TA-Luc, and the control plasmid pRL-TK were transfected into HEK293T cells. Ratios of firefly luciferase activities to Renilla luciferase activities obtained in triplicated wells are shown as the means ± SEMs. (A) pCAGGS (Vec), pCAGGS-HAUL42WT (WT), -HAUL42PA (PA), -HAUL42ΔN (ΔN), and -HAUL42ΔI (ΔI). (B) pEGFP-C1 (Vec), -UL42WT (WT), -UL42AY (AY), and -UL42Ct (Ct).
(PPTX)

**S2 Fig. Intracellular localization of UL42 derivatives.** HA-tagged UL42 proteins (A) and EGFP-tagged UL42 proteins (B) were expressed in HEK293T cells and analyzed with immunofluorescence assay. A. The cells expressing HA-tagged UL42 derivatives were stained with anti-HA (red) and anti-c-Jun (green) antibodies. The nuclei were stained with DAPI (blue). B. The cells expressing EGFP-tagged UL42 derivatives were reacted with anti c-Jun antibody and then with Alexa Flora 647-conjugated secondary antibody (red). EGFP fluorescence and nuclei staining with DAPI are shown in green and blue, respectively. Bar = 10μm.
(PPTX)

**S3 Fig. The phosphorylation status of c-Jun in fibroblasts infected with HCMV wild-type or UL42 mutants.** Fibroblasts, hTERT-BJ1 cell- were mock infected (m) or infected with the following HCMV strains at a multiplicity of infection (MOI) of 5, harvested at 3 day post-infection, and their lysates were analyzed by immunoblotting with the indicated antibodies. WT: HCMV encoding wild-type UL42, R: HCMV encoding rescued UL42, PA: HCMV encoding UL42PA, Δ: HCMV lacking UL42.
(PPTX)

## Author Contributions

**Conceptualization:** Tetsuo Koshizuka.

**Data curation:** Tetsuo Koshizuka.

**Funding acquisition:** Tetsuo Koshizuka.

**Investigation:** Tetsuo Koshizuka.

**Project administration:** Tetsuo Koshizuka.

**Supervision:** Tetsuo Koshizuka, Naoki Inoue.

**Writing – original draft:** Tetsuo Koshizuka.

**Writing – review & editing:** Tetsuo Koshizuka, Naoki Inoue.

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
