## [Decision Letter · Decision Letter 0]

6 Feb 2020

PONE-D-20-02197

Activation of c-Jun by human cytomegalovirus UL42 through JNK activation.

PLOS ONE

Dear Dr. Koshizuka,

Thank you for submitting your manuscript to PLOS ONE. After careful consideration, we feel that it has merit but does not fully meet PLOS ONE’s publication criteria as it currently stands. Therefore, we invite you to submit a revised version of the manuscript that addresses the points raised during the review process. More specifically, the three experts who reviewed your study identified a number of problems in content and presentation that need to be addressed.

We would appreciate receiving your revised manuscript by Mar 22 2020 11:59PM. To enhance the reproducibility of your results, we recommend that if applicable you deposit your laboratory protocols in protocols.io, where a protocol can be assigned its own identifier (DOI) such that it can be cited independently in the future. For instructions see: http://journals.plos.org/plosone/s/submission-guidelines#loc-laboratory-protocols

We look forward to receiving your revised manuscript.

Kind regards,

Michael Nevels

Academic Editor

PLOS ONE

Journal Requirements:

1. PLOS ONE now requires that authors provide the original uncropped and unadjusted images underlying all blot or gel results reported in a submission’s figures or Supporting Information files. This policy and the journal’s other requirements for blot/gel reporting and figure preparation are described in detail at https://journals.plos.org/plosone/s/figures#loc-blot-and-gel-reporting-requirements and https://journals.plos.org/plosone/s/figures#loc-preparing-figures-from-image-files. When you submit your revised manuscript, please ensure that your figures adhere fully to these guidelines and provide the original underlying images for all blot or gel data reported in your submission. See the following link for instructions on providing the original image data: https://journals.plos.org/plosone/s/figures#loc-original-images-for-blots-and-gels.

Reviewers' comments:

Reviewer's Responses to Questions

**Comments to the Author**

1. Is the manuscript technically sound, and do the data support the conclusions?

Reviewer #1: Partly

Reviewer #2: No

Reviewer #3: Partly

2. Has the statistical analysis been performed appropriately and rigorously? 

Reviewer #1: No

Reviewer #2: Yes

Reviewer #3: No

3. Have the authors made all data underlying the findings in their manuscript fully available?

Reviewer #1: Yes

Reviewer #2: Yes

Reviewer #3: Yes

4. Is the manuscript presented in an intelligible fashion and written in standard English?

Reviewer #1: Yes

Reviewer #2: Yes

Reviewer #3: No

5. Review Comments to the Author

Reviewer #1: This manuscript by Koshizuka reported that HCMV protein UL42 activated c-Jun/JNK signaling in an Itch-independent manner. Data showed that different structure domains of UL42 may have different functions, as the internal region of UL42 is important for AP-1 transcriptional activation and the c-terminal region of UL42 is necessary for protein stability. It is an incremental yet worthwhile advance in our understanding. However, most experiments were performed in 293T cells by overexpressing UL42 in isolation. It is important to test whether UL42 is able to activate JNK during CMV infection. My specific comments are as follows:

1. Figure 1,

1) The reporter plasmid with pAP1 needs a detailed description

2) Also, the PPXY to PPXA alternation needs be clarified.

3) Why UL42Ct start from 99aa ratherthan 87aa in the EGFP-UL42Ct mutant?

2. Figure 2A,

1) The way of tagging HA in mutants needs more detailed description.

2) Why the signal intensity of c-Jun is different between vector, △Ct and WT? As is shown in Figure 3 its expression level is similar among the groups. A ratio between nuclear and cytoplasm c-Jun signal may be more convincing, and a statistic analysis is necessary.

3. Figure 2B, it is necessary to add a positive group “EGFP-ul42wt” to rule out the possibility that EGFP has influence on intracellular localization of c-Jun.

4. Line 100-103. The conclusion”These results indicate that the C-terminal region of UL42, but not the PY motif, is involved in the activation of AP-1-dependent transcription” may not be accurate, as shown in figure 3, C-terminal domain of UL42 is important for protein stability, so the low luciferase activity may be because of low protein expression level.

5. Line113-114. The conclusion about C-terminal domain of UL42 may also not be accurate, because the protein expression level of UL42 is extremely low.

6. line 93 “;” should be “:”

7. line105 “UL42wt, we” should be “UL42wt. We”

Reviewer #2: The manuscript by Koshizuka and Inoue shows a novel role for the human cytomegalovirus (HCMV) protein UL42, activating AP-1 via c-Jun and JNK phosphorylation. The authors have previously reported that this viral protein interacts with the ubiquitin E3 ligase Itch, degrading it. They now show, using UL42 mutants, that the functional domain employed by this viral protein to regulate c-Jun activation does not map to the two PPXY motifs responsible of the interaction with Itch, but instead it involves a region containing the C-terminal TMD. Overall, the study reports an interesting observation and suggests a potential contribution of UL42 to viral replication via activation c-Jun/JNK.

However, a mayor problem of the study is the minimal amount of data presented, which in addition do not provide information on the mechanism by which UL42 activates c-Jun and JNK. Another mayor issue lies on the fact that all the observations described in the study are based on the UL42 protein ectopically expressed, and there are not indications that these UL42-mediated processes occur in the context of the HCMV infection. Although HCMV encodes additional proteins that activate the JNK pathway, it would seem important to determine the effect of a HCMV defective in UL42, or more specifically of HCMVs with mutant versions of UL42, in the activation of c-Jun/JNK. In particular, since the authors have previously generated and reported on a HCMV defective on UL42. Finally, the authors do not convincingly determine the region involved in c-Jun/JNK activation, as some conclusions are drawn from mutants with vast deletions, and in the case of UL42�Ct, from a protein with stability problems. Therefore, the introduction of more subtle mutations in UL42 are recommended.

Other concerns:

- Figure 2B does not include clear pictures that support the conclusions drawn by the authors in the manuscript. It is intriguing why the UL42Ct has been generated as a GFP fusion protein and not as an HA-tagged-protein in the same way as WT UL42 and the rest of UL42 mutants. UL42Ct should be cloned in the HA-based vector and analyzed in this background so results are clearer and they could be directly compared with those obtained with the rest of constructs.

- The authors should have included the UL42Ct protein in the assays shown in Fig 1 and 3.

- Conclusions have been drawn based on the UL42�Ct mutant. However, Fig 3 shows that the corresponding protein is not expressed in transfected cells, and the authors indicate that this might be due to the instability of the protein. It is hard to know what to make of it. The observation probably questions the results obtained with this mutant. In addition, if this is the case, why then the authors detect the expression of UL42�Ct by immunofluorescence in Fig. 2A?

- In order to base conclusions from Fig 2A, images of some of the most relevant immunofluorescence pictures at a higher magnification should have been added.

Minor concerns:

- The Material and Methods miss information. For example, it does not contain immunoblot or fluorescence microscopy sections that could provide details on the conditions used in these procedures, such as lysis buffer, percentage of SDS-PAGE, permeabilizing conditions, type of microscopy (is it confocal?), or magnification used in the study.

- The authors have previously reported that UL42 degrades Itch via the two PPXY motifs. However, and although they do not present a quantification of the normalized Itch band, this does not seem the be the case in the western blot shown in Figure 3. They should comment on this.

- Why in the western blot shown in Fig 3 UL42�N migrates as two bands?

- The information of the function recently attributed to UL42 (Fu et al. PLoS Pathogens 2019, 15(5): e1007691) should be mentioned in the Introduction too to contextualize the viral protein.

Reviewer #3: Is the manuscript technically sound, and do the data support the conclusions?

Koshizuka et. al, present evidence herein that the human cytomegalovirus protein UL42 activates c-Jun via JNK in HEK 293T cells. Although the work is technically sound in most places, revisions must be made before the conclusions drawn by the authors are fully supported by the data. One area of concern is Figure 1B, where it is stated that “ratios… obtained in triplicated wells are shown as the mean ±SEM” suggesting that data is from a single experimental repeat. Can this result be reproduced and could the authors show these results as the mean ±SEM from across multiple experimental replicates?

In Figure 2, the authors show that their EGFP-UL42Ct construct does not induce nuclear localization of c-Jun. This is a very interesting and important result. Could the authors therefore also show that this construct does not activate Jun/JNK/AP-1 either by luciferase assay as in figure 1, and/or by immunoblot as in figure 3? This is not vital for the integrity of the manuscript but would greatly enhance the claims being made.

In Figure 3, the authors were unable to detect UL42∆Ct by immunoblot and conclude that UL42∆Ct is too unstable to be detected. This raises problems with their results in Figure 1, as they do not demonstrate that UL42∆Ct was expressed by the cells in this assay. If UL42∆Ct was not being expressed, then the authors cannot conclude that the C terminal TMD is required for UL42-mediated activation of AP-1...

...However, in Figure 2A the authors can detect UL42∆Ct by immunofluorescence assay. Figure 2A therefore suggests that UL42∆Ct is expressed just as well as other UL42 mutants. This indicates that there is potentially a problem with the immunoblotting protocol that is used to solubilize and detect UL42 in Figure 3. Since UL42∆Ct is no longer a membrane protein, it could be that a different immunoblotting approach is required to detect it. It is difficult to suggest what other problems may be causing this as the authors have not included a detailed explanation of their immunoblotting protocol in the methods section.

Figure 2 provides strong evidence to support the conclusions drawn by the authors. I believe that the improvements to the data in Figures 1 and 3 that I have stated so far would strengthen the rigor of this manuscript sufficiently to justify the author’s conclusions.

Has the statistical analysis been performed appropriately and rigorously?

Statistical analysis, ideally using one-way analysis of variance (ANOVA), needs to be performed for luciferase data in Figure 1B in order to calculate the statistical significance of the results presented. This analysis should be performed on data from three experimental repeats.

Is the manuscript presented in an intelligible fashion and written in standard English?

For a reader without expertise in this field, areas of the manuscript require further details to be intelligible. For example:

• The abbreviations (PA) and (∆N) are not defined in the text.

• A control vector is used, but it is labeled as “V” in Figure 1, “Vector” in Figure 2 and “Vec” in Figure 3. The authors should state if these are the same vector, and what this vector is (I assumed that it is the empty UL42 expression vector, pCAGGS).

• Is anything already known about the internal region or C terminal TMD? If so, this should be detailed in the introduction (or stated that nothing is known).

• When the authors discuss the proximal regions of the C-terminal TMD, is this the same region as the internal region, or are they different? The authors could label Figure 1A to help clarify which regions of UL42 they are discussing.

• The figure legends for figures 1 and 2 need to include definitions for the UL42 mutants, as in figure 3.

There is no mention in the methods section of how immunoblots were performed, a detail which is important for understanding the lack of expression of the ∆Ct mutant in Figure 3. Other details that should be included in the methods section:

• Antibody concentrations for immunofluorescence and immunoblotting

• Antibody incubation conditions

• Enzymes and ligases used for cloning UL42 into the pCAGGs vector would be useful

• The amount of each plasmid that was transfected for each assay

6. PLOS authors have the option to publish the peer review history of their article (what does this mean?). If published, this will include your full peer review and any attached files.

Reviewer #1: No

Reviewer #2: No

Reviewer #3: Yes: Dr Benjamin Anthony Krishna

---

## [Author Response · Author response to Decision Letter 0]

19 Mar 2020

Responses to Reviewer’s comments

MS#:PONE-D-20-02197

We thank you for your review on our manuscript. To accommodate your comments, we added a couple of experiments and modified the text, figures and supplementary materials as described below. Line numbers indicated by the reviewers are based on the original manuscript and those by us are based on the revised manuscript with tracked changes (file: PONE-D-20-02197 Revised Manuscript with Track Changes). 

Reviewer #1

General comment: It is important to test whether UL42 is able to activate JNK during CMV infection.

Response:　To accommodate the comment, we conducted an additional experiment using recombinant HCMV strains with mutated versions of UL42 and found that HCMV infection induced c-Jun phosphorylation although the deletion of UL42 did not affect the c-Jun phosphorylation status in HCMV infected cells. These results could be explained by the presence of multiple mechanisms of the c-Jun phosphorylation in HCMV-infected cells. To describe our observation and discussion on this observation, we added immunoblot images as Figure S2 as well as sentences describing construction of recombinant HCMV strains (lines 143- 156), the results (lines 228-230) and discussion (lines 259- 261).

Comment 1 (Figure 1): The reporter plasmid with pAP1 needs a detailed description. Also, the PPXY to PPXA alternation needs be clarified. Why UL42Ct start from 99aa rather than 87aa in the EGFP-UL42Ct mutant?

Response: We used a commercial reporter plasmid, pAP1(PMA)-TA-Luc, which contains multiple copies of the AP1 element and a minimal promoter, located upstream of the firefly luciferase gene. Descriptions of these details were added into the text (lines 98- 102).

 Construction of the UL42PA mutant was described in our previous study (Koshizuka et al., 2016 J. Gen. Virol.).　 Briefly, tyrosine (Y) codons in the two PPXY (PY) motifs were replaced with alanine (A) codons by site-directed mutagenesis, resulting in the two PPXA (PA) motifs.　 We modified the text for clarification (lines 175-179). 

 Thanks for pointing out the error of the UL42Ct start site. EGPF-UL42Ct should contain amino acid residues from 87 to 124. Accordingly, Figure 1A was corrected. 

Comment 2 (Figure 2A): The way of tagging HA in mutants needs more detailed description. Why the signal intensity of c-Jun is different between vector, △Ct and WT? As is shown in Figure 3 its expression level is similar among the groups. A ratio between nuclear and cytoplasm c-Jun signal may be more convincing, and a statistic analysis is necessary.

Response: The details of construction of HA-tagged UL42 plasmids were added to the Materials and Methods section (lines 77-83). 

 As the reviewer noticed, although the amounts of c-Jun in the cells were at a similar level in immunoblotting, the signal intensities of c-Jun were different in spite of the use of the same anti-c-Jun antibody in both assays. We assume that the used antibody reacted better for nuclear c-Jun.

 To accommodate the comment, the percentages of cells containing nuclear c-Jun among the cells expressing UL42WT or mutated versions were obtained for statistical analyses. Descriptions for the procedures and for the statistical results were added to the Materials and Methods section (lines 158-168) and to the Results section (lines 198-201 and 210-212), respectively. The statistically significances are also shown in the revised Figures 1B, 1C, 2B and 3B.

Comment 3: Figure 2B, it is necessary to add a positive group “EGFP-UL42wt” to rule out the possibility that EGFP has influence on intracellular localization of c-Jun.

Response: As suggested by the reviewer, we conducted an experiment that included EGFP-UL42WT and -UL42AY as controls (Figures 3 and 4 in the revised manuscript). Both EGFP-UL42WT and -UL42AY induced AP1 signaling and c-Jun nuclear accumulation and phosphorylation, which is consistent with the results using HA-tagged UL42 derivatives. To describe those results, the text was modified (lines 84-89, 181-185, 202-213, and 226-227).

Comment 4: Line 100-103. The conclusion ”These results indicate that the C-terminal region of UL42, but not the PY motif, is involved in the activation of AP-1-dependent transcription” may not be accurate, as shown in figure 3, C-terminal domain of UL42 is important for protein stability, so the low luciferase activity may be because of low protein expression level. Line113-114. The conclusion about C-terminal domain of UL42 may also not be accurate, because the protein expression level of UL42 is extremely low.

Response: In an additional experiment using EGFP-fusion proteins (revised Fig. 1C and Fig. 3), we found that the C-terminal transmembrane domain of UL42 alone did not accumulate c-Jun in the nuclei nor activated c-Jun. As criticized by the reviewer, the amounts of HA-tagged UL42ΔCt protein fluctuated from one experiment to another due to the stability issue. Taking account of the additional results and the stability issue, we removed the results obtained by expression of HA-tagged UL42ΔCt and modified our conclusion. 

Minor comment1: line 93 “;” should be “:”

Minor comment 2: line105 “UL42wt, we” should be “UL42wt. We”

Response: As suggested, we corrected those errors (lines 114-118, and 181).

Reviewer #2

General comment 1: 

However, a major problem of the study is the minimal amount of data presented, which in addition do not provide information on the mechanism by which UL42 activates c-Jun and JNK. Another major issue lies on the fact that all the observations described in the study are based on the UL42 protein ectopically expressed, and there are not indications that these UL42-mediated processes occur in the context of the HCMV infection. Although HCMV encodes additional proteins that activate the JNK pathway, it would seem important to determine the effect of a HCMV defective in UL42, or more specifically of HCMVs with mutant versions of UL42, in the activation of c-Jun/JNK. In particular, since the authors have previously generated and reported on a HCMV defective on UL42. Finally, the authors do not convincingly determine the region involved in c-Jun/JNK activation, as some conclusions are drawn from mutants with vast deletions, and in the case of UL42ΔCt, from a protein with stability problems. Therefore, the introduction of more subtle mutations in UL42 are recommended.

Response: As described in the response to Reviewer #1’s General comment, we conducted an additional experiment using recombinant HCMV strains and found that the lack of UL42 did not affect the c-Jun phosphorylation status in HCMV-infected cells. These results could be explained by the presence of multiple mechanisms of the c-Jun phosphorylation in HCMV-infected cells. We added the immunoblot images as Figure S2 and modified the text (lines 143-156, and 259-261). 

 In addition, we conducted addition experiments using plasmids expressing EGFP-tagged UL42 derivatives and found that both EGFP-UL42WT and -UL42AY activated AP-1 signaling. These observations were consistent with the results using the plasmids expressing HA-tagged proteins. At the same time, AP-1 activation, nuclear accumulation of c-Jun and c-Jun phosphorylation were significantly reduced in the cells expressing EGFP-UL42Ct. Therefore, it is likely that the C-terminal TMD was not involved in AP-1 signaling. We modified the text (lines 84-89, 181-185, 202-213, and 226-227) and revised Fig. 1C and Fig. 3. As criticized by the reviewer, the amounts of HA-tagged UL42ΔCt protein fluctuated from one experiment to another due to the stability issue. Taking account of the additional results and the stability issue, we removed the results obtained by expression of HA-tagged UL42ΔCt, and modified our conclusion. 

 As we agree with the need of experiments using more subtle mutations, we would like to address such a fine mapping in the next step. We added a description regarding this issue as one of limitations of this study (lines 257-258).

Comment 1: Figure 2B does not include clear pictures that support the conclusions drawn by the authors in the manuscript. It is intriguing why the UL42Ct has been generated as a GFP fusion protein and not as an HA-tagged-protein in the same way as WT UL42 and the rest of UL42 mutants. UL42Ct should be cloned in the HA-based vector and analyzed in this background so results are clearer and they could be directly compared with those obtained with the rest of constructs.

Response: To make UL42Ct data comparable with HA-tagged UL42 derivatives, we added a plasmid expressing EGFP-UL42WT as a positive control and a plasmid expressing UL42AY as a PPxY mutant in the revised Fig. 3. In addition, to show clear subcellular localization, the data of immunofluorescence assay at a higher magnification were added as Figure S1. Accordingly, the text was modified (lines 84-89, 181-185, 202-213, and 226-227)

Comment 2: The authors should have included the UL42Ct protein in the assays shown in Fig 1 and 3.

Comment 3: Conclusions have been drawn based on the UL42ΔCt mutant. However, Fig 3 shows that the corresponding protein is not expressed in transfected cells, and the authors indicate that this might be due to the instability of the protein. It is hard to know what to make of it. The observation probably questions the results obtained with this mutant. In addition, if this is the case, why then the authors detect the expression of UL42ΔCt by immunofluorescence in Fig. 2A?

Response: As described for Comment 1, we added EGFP-UL42WT and UL42AY to compare with EGFP-UL42Ct and the C-terminal TMD of UL42 was not involved in AP-1 activation. Taking account of the new results, we modified our conclusion.

Comment 4: In order to base conclusions from Fig 2A, images of some of the most relevant immunofluorescence pictures at a higher magnification should have been added.

Response: As suggested, we added the immunofluorescence assay images at a higher magnification as Figure S1.

Minor concerns:

Comment 5: The Material and Methods miss information. For example, it does not contain immunoblot or fluorescence microscopy sections that could provide details on the conditions used in these procedures, such as lysis buffer, percentage of SDS-PAGE, permeabilizing conditions, type of microscopy (is it confocal?), or magnification used in the study.

Response: As suggested by the reviewer, we intensively modified the Materials and Methods section (lines 67-168). 

Comment 6: The authors have previously reported that UL42 degrades Itch via the two PPXY motifs. However, and although they do not present a quantification of the normalized Itch band, this does not seem to be the case in the western blot shown in Figure 3. They should comment on this.

Response: In order to show more clear results, the immunoblot image of Itch in Figure 4A was revised. As shown in Figure 4A, Itch was decreased in UL42WT and ΔI expressing cells but not in UL42PA or ΔN expressing cells. 

Comment 6: Why in the western blot shown in Fig 3 UL42ΔN migrates as two bands?

Response: Based on the molecular weights of the two bands, we assume that they are products of proteolytic cleavage (lines 218-219).

Comment 7: The information of the function recently attributed to UL42 (Fu et al. PLoS Pathogens 2019, 15(5): e1007691) should be mentioned in the Introduction too to contextualize the viral protein. 

Response: As suggested by the reviewer, we described the study in the Introduction section (lines 56 – 59).

Reviewer #3

General comment: Although the work is technically sound in most places, revisions must be made before the conclusions drawn by the authors are fully supported by the data. One area of concern is Figure 1B, where it is stated that “ratios… obtained in triplicated wells are shown as the mean ±SEM” suggesting that data is from a single experimental repeat. Can this result be reproduced and could the authors show these results as the mean ±SEM from across multiple experimental replicates?

Response: Our luciferase assay results are based on three independent experiments and each set of experimental results was analyzed with the one-way ANOVA test followed by the Tukey’s comparison test using GraphPad PRISM software. The data obtained in triplicated wells of one of three independent experiments are presented as Figure 1B and C. The text were revised for clarification (lines 159-168 and 372-380).

Comment 1: In Figure 2, the authors show that their EGFP-UL42Ct construct does not induce nuclear localization of c-Jun. This is a very interesting and important result. Could the authors therefore also show that this construct does not activate Jun/JNK/AP-1 either by luciferase assay as in figure 1, and/or by immunoblot as in figure 3? This is not vital for the integrity of the manuscript but would greatly enhance the claims being made.

Response: To accommodate this comment, EGFP-UL42WT and UL42AY were added in this manuscript as controls for EGFP-UL42Ct. AP-1 signal was activated and c-Jun was accumulated to nucleus by expression of EGFP-UL42WT and -UL42AY but not by EGFP-UL42Ct. To describe these results, the text was modified (lines 84-89, 181-185, 202-213, and 226-227) and revised Fig. 1C and Fig. 3.

Comment 2: In Figure 3, the authors were unable to detect UL42∆Ct by immunoblot and conclude that UL42∆Ct is too unstable to be detected. This raises problems with their results in Figure 1, as they do not demonstrate that UL42∆Ct was expressed by the cells in this assay. If UL42∆Ct was not being expressed, then the authors cannot conclude that the C terminal TMD is required for UL42-mediated activation of AP-1.

Comment 3: However, in Figure 2A the authors can detect UL42∆Ct by immunofluorescence assay. Figure 2A therefore suggests that UL42∆Ct is expressed just as well as other UL42 mutants. This indicates that there is potentially a problem with the immunoblotting protocol that is used to solubilize and detect UL42 in Figure 3. Since UL42∆Ct is no longer a membrane protein, it could be that a different immunoblotting approach is required to detect it. It is difficult to suggest what other problems may be causing this as the authors have not included a detailed explanation of their immunoblotting protocol in the methods section.

Response: As described for the comments of Reviewers #1 and #2, we conducted addition experiments using plasmids expressing EGFP-fusion proteins and found that both EGFP-UL42WT and -UL42AY activated AP-1 signaling. These observations were consistent with the results using the plasmids expressing HA-tagged proteins. As in the same experiments, AP-1 activation and c-Jun phosphorylation were significantly reduced in the cells expressing EGFP-UL42Ct, it is likely that the C-terminal TMD was not involved in AP-1 signaling. We modified the text (lines 84-89, 181-185, 202-213, and 226-227) and revised Fig. 1C and Fig. 3. As criticized by the reviewer, the amounts of HA-tagged UL42ΔCt protein fluctuated from one experiment to another due to the stability issue. Taking account of the additional results and the stability issue, we removed the results obtained by expression of HA-tagged UL42ΔCt and modified our conclusion. 

Comment 4: Figure 2 provides strong evidence to support the conclusions drawn by the authors. I believe that the improvements to the data in Figures 1 and 3 that I have stated so far would strengthen the rigor of this manuscript sufficiently to justify the author’s conclusions.

Response: To accommodate the comment, we added the statistical analysis in Figures 1 and 2. In addition, we added EGFP-UL42WT and -UL42AY to compare EGFP-UL42Ct (revised Fig. 3). 

Comment 5: Has the statistical analysis been performed appropriately and rigorously?

Comment 6: Statistical analysis, ideally using one-way analysis of variance (ANOVA), needs to be performed for luciferase data in Figure 1B in order to calculate the statistical significance of the results presented. This analysis should be performed on data from three experimental repeats.

Response: As suggested, we conducted statistical analyses using GlaphPad PRISM software. All statistical analysis was performed on data from three independent experiments and we confirmed that these data were significant statistically. The text was revised (lines 158-168, 374-378, 388-392, and 398-401).

Comment 7: Is the manuscript presented in an intelligible fashion and written in standard English?

Response: Our manuscript was edited commercially by an English-native scientific editor.

Comment 8-1: The abbreviations (PA) and (∆N) are not defined in the text.

Response: Sentences describing the definitions of PA and ∆N were added to the text (lines 79-83, and 179-181).

Comment 8-2: A control vector is used, but it is labeled as “V” in Figure 1, “Vector” in Figure 2 and “Vec” in Figure 3. The authors should state if these are the same vector, and what this vector is (I assumed that it is the empty UL42 expression vector, pCAGGS).

Response: We modified the legends for these figures.

Comment 8-3: Is anything already known about the internal region or C terminal TMD? If so, this should be detailed in the introduction (or stated that nothing is known).

Response: As there are no additional information about the domain mapping of UL42, the text was revised for clarification (lines 49-52).

Comment 8-4: When the authors discuss the proximal regions of the C-terminal TMD, is this the same region as the internal region, or are they different? The authors could label Figure 1A to help clarify which regions of UL42 they are discussing.

Response: For clarification, the sentence was revised (line 212-213, 246-249, and 275-276).

Comment 8-5: The figure legends for figures 1 and 2 need to include definitions for the UL42 mutants, as in figure 3.

Response: The legends were revised (lines 378-380, 383-387, and 398-400).

Comment 9: There is no mention in the methods section of how immunoblots were performed, a detail which is important for understanding the lack of expression of the ∆Ct mutant in Figure 3. Other details that should be included in the methods section:

• Antibody concentrations for immunofluorescence and immunoblotting

• Antibody incubation conditions

• Enzymes and ligases used for cloning UL42 into the pCAGGS vector would be useful

• The amount of each plasmid that was transfected for each assay

Response: The Materials and Methods section were intensively revised to include these information. (lines 76-95, 97-104, 106-118, and 120-141)

Others

1. To increase readability, the text was edited without losing the original contents.

2. Primers names were simplified.

---

## [Decision Letter · Decision Letter 1]

2 Apr 2020

PONE-D-20-02197R1

Activation of c-Jun by human cytomegalovirus UL42 through JNK activation.

PLOS ONE

Dear Dr. Koshizuka,

Thank you for submitting your revised manuscript to PLOS ONE. Please address the remaining minor issues two of the referees have raised.

We would appreciate receiving your revised manuscript by May 17 2020 11:59PM. To enhance the reproducibility of your results, we recommend that if applicable you deposit your laboratory protocols in protocols.io, where a protocol can be assigned its own identifier (DOI) such that it can be cited independently in the future. For instructions see: http://journals.plos.org/plosone/s/submission-guidelines#loc-laboratory-protocols

We look forward to receiving your revised manuscript.

Kind regards,

Michael Nevels

Academic Editor

PLOS ONE

Reviewers' comments:

Reviewer's Responses to Questions

**Comments to the Author**

1. If the authors have adequately addressed your comments raised in a previous round of review and you feel that this manuscript is now acceptable for publication, you may indicate that here to bypass the “Comments to the Author” section, enter your conflict of interest statement in the “Confidential to Editor” section, and submit your "Accept" recommendation.

Reviewer #1: (No Response)

Reviewer #2: All comments have been addressed

Reviewer #3: (No Response)

2. Is the manuscript technically sound, and do the data support the conclusions?

Reviewer #1: Yes

Reviewer #2: Yes

Reviewer #3: Yes

3. Has the statistical analysis been performed appropriately and rigorously? 

Reviewer #1: Yes

Reviewer #2: Yes

Reviewer #3: Yes

4. Have the authors made all data underlying the findings in their manuscript fully available?

Reviewer #1: Yes

Reviewer #2: Yes

Reviewer #3: Yes

5. Is the manuscript presented in an intelligible fashion and written in standard English?

Reviewer #1: Yes

Reviewer #2: Yes

Reviewer #3: Yes

6. Review Comments to the Author

Reviewer #1: This manuscript is a revised version of one I previously reviewed. It is much better that the authors have addressed most of my comments. However, there are still some problems need to be corrected

Minor issues: line247-248 “c-Jun” should be “C-Jun”

Line246-247 needs rephrasing.

Line184-185, this sentence is confusing, according to the results, “except for” should be “rather than”

Comment 1: Figure 1. It is good to use HA and GFP to rule out the effects of tag protein on results. But the experiments control from this two different tag groups should keep consistent.

Tagged-UL42WT, -UL42N, -UL42I, -UL42PY, -UL42AY, and -CT should be included in both two groups.

Comment 2: Line226-227. ”c-Jun and JNK were phosphorylated by the expression of EGFP-UL42WT and -UL42AY but not by EGFP-UL42Ct.” But as Figure 4B shows, the phosphorylation level of c-Jun by EGFP-UL42Ct is almost the same as EGFP-UL42WT and UL42AY and the phosphorylation level of JNK by EGFP-UL42Ct is nearly to EGFP-UL42AY.

More experiment controls need to be set to explain this problem or the conclusion need to be amended.

Comment 3 : Line 275-276. As comment 2 mentioned, taken all results into consideration, aa52-86 region of UL42 is just responsible for Jun translocation to nucleus.

Reviewer #2: The authors have addressed most of the concerns previously raised. I beleive that the manuscript is now suitable for publication.

The legend to Fig S2 needs to be revised. It is not correct to state "Fibroblasts infected with mock (m) ..."

Reviewer #3: I would like to thank Koshizuka and Inoue for the changes they have made to their manuscript, which shows that the HCMV gene UL42 activates Jun in transfected HEK293T cells, and that the internal region is important for this activity. As suggested, they have better defined their materials and constructs, which helps the reader to understand the manuscript. Could the authors make the following changes to their figures to improve reader understanding:

-Figure 1B shows the results from one of three experiments, including statistical analysis. Could the authors include the data for the other two experiments, either as mean averages in Figure 1B, or at least in the supplemental section as two separate figures?

-In Figure 2B, could the authors include the empty vector control in their quantification of Jun positive nuclei? Could the authors also label the y axis to describe what this graph shows? Similarly for Figure 3B, the y axis needs a label.

-Figure S1A requires an empty vector control.

Some changes also need to be made to the text:

-The sentence in results section lines 181-185 does not make sense. I think that removing the word "and" so that this reads: "...the PY motifs is essential for the activation of AP-1 dependent transcription" is what the authors want to say.

- In figures 3A and 4B, EGFP-Ct is detectable by immunofluorescence but not by western blot. The authors suggest that the difference in their observation is due to variable protein stability. Having examined figures 3A and S1B, it does appear as if the EGFP-UL42Ct construct has weaker fluorescence than the other constructs. Would the authors agree with this, as it would then make their results consistent? If so, could the authors mention this weaker fluorescence in the discussion.

- The final paragraph of the results section describes the infection assay performed by the authors. Could the authors include details such as cell type infected, multiplicity of infection and the duration of infection? This information is vital to analyzing this data and should be mentioned in the results as well as the figure legend. Perhaps, for example, if the authors have infected these cells at a higher MOI (higher than 1) this could lead to significant activation of Jun simply due to the large amount of virus present. Using a lower MOI may therefore reveal differences between WT and UL42 deletion viruses for Jun activation. Without this information, it is hard to determine the validity of this result.

- It is not surprising that there was no difference between WT and the UL42 deletion mutant for Jun activation, given that other HCMV genes also activate Jun, as stated by the authors. As I understand it, Dunn et al showed in 2003 that UL42 is dispensable for infection in fibroblasts. This should be mentioned by the authors as it supports the notion that UL42's function is redundant during infection.

- As the transfection assays were performed in HEK293T cells, and the infections performed in fibroblasts, could activation of Jun by UL42 be cell-type specific? The authors could either perform transient transfection assays in fibroblasts to resolve this difference, or simply moderate their discussion section to specify that this observation is in transfection assays of HEK293T cells and not fibroblasts.

7. PLOS authors have the option to publish the peer review history of their article (what does this mean?). If published, this will include your full peer review and any attached files.

Reviewer #1: No

Reviewer #2: No

Reviewer #3: Yes: Benjamin Krishna

---

## [Author Response · Author response to Decision Letter 1]

17 Apr 2020

Responses to Reviewer’s comments

MS#:PONE-D-20-02197R1

We thank you for your review on the revised version of our manuscript. To accommodate your comments, we modified the text and supplementary materials as described below. Line numbers indicated by the reviewers are based on the revised manuscript and those by us are based on the re-revised manuscript with tracked changes (file: PONE-D-20-02197R1 Revised Manuscript with Track Changes). 

Reviewer #1

Comment 1 (Figure 1): It is good to use HA and GFP to rule out the effects of tag protein on results. But the experiments control from this two different tag groups should keep consistent. Tagged-UL42WT, -UL42N, -UL42I, -UL42PY, -UL42AY, and -CT should be included in both two groups.

Response: We used EGFP-UL42Ct to confirm the requirement of the UL42 C-terminus region for activation of c-Jun/JNK signaling. On the other hand, the EGFP-UL42AY mutant was used for a separate issue, that is, confirmation of no involvement of the Nedd4 family E3 ligases in the c-Jun/JNK signaling. The text was revised to clarify the respective purposes of these constructs (lines 181-185).

Comment 2: Line226-227. ”c-Jun and JNK were phosphorylated by the expression of EGFP-UL42WT and -UL42AY but not by EGFP-UL42Ct.” But as Figure 4B shows, the phosphorylation level of c-Jun by EGFP-UL42Ct is almost the same as EGFP-UL42WT and UL42AY and the ph9osphorylation level of JNK by EGFP-UL42Ct is nearly to EGFP-UL42AY. More experiment controls need to be set to explain this problem or the conclusion need to be amended.

Response: Although the phosphorylation level of c-Jun seems weak in EGFP-UL42WT and -AY expressing cells, the phosphorylation of JNK was significantly increased in EGFP-UP42WT or -AY expressing cells but not in EGFP-UL42Ct expressing cells (Fig.4B). In addition, the results of luciferase assay (Fig.1C) and nuclear translocation of c-Jun (Fig.3) supported the notion that the phosphorylation of c-Jun and activation of AP-1 signaling occurred by expression of EGFP-UL42WT or -AY but not by expression of EGFP-UL42Ct. To accommodate the reviewer’s comment, we added some sentences discussing the issue (lines 246-253)

Comment 3 : Line 275-276. As comment 2 mentioned, taken all results into consideration, aa52-86 region of UL42 is just responsible for Jun translocation to nucleus.

Response: We agree. To accommodate the comment, we modified the text (lines 298-299) 

Minor comments: 

1. line247-248 “c-Jun” should be “C-Jun”

2. Line246-247 needs rephrasing.

3. Line184-185, this sentence is confusing, according to the results, “except for” should be “rather than”

Response: As suggested, these sentences were corrected or rephrased (lines 196, 202, and 230 for #1, lines 256-259 for #2, and lines 185-189 for #3). 

Reviewer #2

Minor comment 1:

The legend to Fig S2 needs to be revised. It is not correct to state "Fibroblasts infected with mock (m) ..."

Response: As suggested, the sentence was corrected (lines 465-471). 

Reviewer #3 

Comment 1: Figure 1B shows the results from one of three experiments, including statistical analysis. Could the authors include the data for the other two experiments, either as mean averages in Figure 1B, or at least in the supplemental section as two separate figures?

Response: As suggested, we revised the sentences describing that the results of Figure 1B and 1C indicated the means of three independent experiments. In addition, we added a supplementary figure (S1 Figure) showing the three independent sets of results (lines 103-105 and 401-404, and 445-453).

Comment 2: In Figure 2B, could the authors include the empty vector control in their quantification of Jun positive nuclei? Could the authors also label the y axis to describe what this graph shows? Similarly for Figure 3B, the y axis needs a label.

Comment 3: Figure S1A requires an empty vector control.

Response: Figure 2B presented the percentage of nuclear c-Jun positive cells in cells expressing HA-UL42 or its variant. It is technically hard to use the empty vector (pCAGGS) for the quantification, as it does not express detectable HA-tagged protein, which is a situation different from that of the empty vector for EGFP fusion. In the same context, it is impossible to add an image with an empty vector control in Figure S1A, as we have no way to distinguish cells transfected with the empty vector from those not transfected. As suggested, the labels of y axis were added to Figures 2B and 3B.

Minor comment 1: The sentence in results section lines 181-185 does not make sense. I think that removing the word "and" so that this reads: "...the PY motifs is essential for the activation of AP-1 dependent transcription" is what the authors want to say.

Response: As suggested, we corrected the sentence (lines 185-189).

Minor comment 2: In figures 3A and 4B, EGFP-Ct is detectable by immunofluorescence but not by western blot. The authors suggest that the difference in their observation is due to variable protein stability. Having examined figures 3A and S1B, it does appear as if the EGFP-UL42Ct construct has weaker fluorescence than the other constructs. Would the authors agree with this, as it would then make their results consistent? If so, could the authors mention this weaker fluorescence in the discussion.

Response: We agreed the reviewer’s comment. We pointed out the weaker signals in the Result section and discussed the issue (lines 260-268).

Minor comment 3: The final paragraph of the results section describes the infection assay performed by the authors. Could the authors include details such as cell type infected, multiplicity of infection and the duration of infection? This information is vital to analyzing this data and should be mentioned in the results as well as the figure legend. Perhaps, for example, if the authors have infected these cells at a higher MOI (higher than 1) this could lead to significant activation of Jun simply due to the large amount of virus present. Using a lower MOI may therefore reveal differences between WT and UL42 deletion viruses for Jun activation. Without this information, it is hard to determine the validity of this result.

Response: As requested, we added the details of infection conditions in the legend for Figure S3 of this revised manuscript (lines 465-471). 

Minor comment 4: It is not surprising that there was no difference between WT and the UL42 deletion mutant for Jun activation, given that other HCMV genes also activate Jun, as stated by the authors. As I understand it, Dunn et al showed in 2003 that UL42 is dispensable for infection in fibroblasts. This should be mentioned by the authors as it supports the notion that UL42's function is redundant during infection.

Response: As suggested, a sentence describing the previous studies with citations, including Dunn et al, was added (lines 290-292). 

Minor comment 5: As the transfection assays were performed in HEK293T cells, and the infections performed in fibroblasts, could activation of Jun by UL42 be cell-type specific? The authors could either perform transient transfection assays in fibroblasts to resolve this difference, or simply moderate their discussion section to specify that this observation is in transfection assays of HEK293T cells and not fibroblasts.

Response: It is very important point, but we did not determine the cell type specificity of the activation of AP-1 signaling by UL42. As the fibroblast cells which we used have very low transfection efficiency, it is difficult to analyze the effects quantitatively. As suggested, we clarified that the observation was in HEK293T cells and addressed the issue in Discussion (lines 276-282). 

Others: 

We rephrased “293T cells” as “HEK293T cells”.

---

## [Editor Report · Decision Letter 2]

20 Apr 2020

Activation of c-Jun by human cytomegalovirus UL42 through JNK activation.

PONE-D-20-02197R2

Dear Dr. Koshizuka,

We are pleased to inform you that your manuscript has been judged scientifically suitable for publication and will be formally accepted for publication once it complies with all outstanding technical requirements.

With kind regards,

Michael Nevels

Academic Editor

PLOS ONE
---

## [Editor Report · Acceptance letter]

24 Apr 2020

PONE-D-20-02197R2 

Activation of c-Jun by human cytomegalovirus UL42 through JNK activation. 

Dear Dr. Koshizuka:

I am pleased to inform you that your manuscript has been deemed suitable for publication in PLOS ONE. Congratulations! Your manuscript is now with our production department. 

With kind regards,

on behalf of

Dr. Michael Nevels 

Academic Editor

PLOS ONE